# Response Surface Methodology in Optimising the Extraction of Polyphenolic Antioxidants from Flower Buds of *Magnolia* × *soulangeana* Soul.-Bod. var. ‘Lennei’ and Their Detailed Qualitative and Quantitative Profiling

**DOI:** 10.3390/molecules28176335

**Published:** 2023-08-30

**Authors:** Grażyna Zgórka, Aldona Adamska-Szewczyk, Tomasz Baj

**Affiliations:** 1Department of Pharmacognosy with the Medicinal Plant Garden, Faculty of Pharmacy, Medical University of Lublin, 1 Chodźki Str., 20-093 Lublin, Poland; tomasz.baj@umlub.pl; 2Bionorica Polska Sp. z o.o., 6B Hrubieszowska Str., 01-209 Warsaw, Poland; a.adamskaszewczyk@gmail.com

**Keywords:** *Magnolia* × *soulangeana* var. ‘Lennei’, response surface methodology, ultrasound-assisted extraction, phytochemical profiling, polyphenolic antioxidants

## Abstract

A response surface methodology (RSM) with a central composite design (CCD) was developed to predict and apply the best ultrasound-assisted extraction (UAE) conditions, including the extraction time, the composition of aqueous-ethanolic extractants, and the solvent-to-plant-material ratio, for obtaining the highest yields of different types of polyphenolic components from the dried flower buds of *Magnolia* × *soulangeana* Soul.-Bod. var. ‘Lennei’ (MSL). The novel approach in the RSM procedure resulted from the simultaneous optimisation of UAE conditions to obtain extracts with the highest antioxidant and antiradical potential (examined as dependent variables), using appropriate spectrophotometric assays, with Folin–Ciocâlteu and 2,2-diphenyl-1-picrylhydrazyl reagents, respectively. The use of 66.8% (*V*/*V*) ethanol as the extraction solvent during the 55.2 min extraction protocol and the ratio of extractant volume to herbal substance of 46.8 mL/g gave the highest total yield of bioactive antioxidant phenolics in the extract obtained. For this herbal preparation, a qualitative and quantitative analysis was performed using combined chromatographic (LC), spectroscopic (PDA), and tandem mass spectrometric (ESI-QToF–MS/MS) techniques. A detailed phytochemical profiling, conducted for the first time, documented substantial amounts of various polyphenolic antioxidants, especially phenylethanoids and flavonoids, in the MSL flower buds. Their average total content exceeded 30.3 and 36.5 mg/g dry weight, respectively.

## 1. Introduction

*Magnolia* × *soulangeana* is a plant hybrid of *M. denudata* Desr. and *M. liliiflora* Desr., which derives its name from French botanist Étienne Soulange-Bodin, who successfully crossed the two aforementioned species in the early 19th century. A large number of hybrid cultivars were later obtained from *M. soulangeana*, among them a variety called ‘Lennei’ with beautiful pear-shaped white and pink flowers (Figure 1). According to The World Flora Online [1], the genus *Magnolia* L. (Magnoliaceae) includes 338 subordinate taxa that are trees or shrubs particularly widely distributed in Southeast Asia (China) and the tropical regions of Southwestern North America and Central America. Unlike most wild magnolias, *M. soulangeana* var. ‘Lennei’, being a hybrid species developed in Europe, is particularly common on this continent (except in Scandinavia) as an ornamental tree in parks and gardens that tolerates low winter temperatures relatively well, so it is popular in cultivation.

Currently, considering biological activity and approved medical uses, *M. officinalis* Rehder et E.H. Wilson and *M. biondii* Pamp. are the best-known magnolia taxa in Europe. The dried bark and/or flowers of these two species are described in separate monographs of the European Pharmacopoeia as herbal substances standardised for the content of characteristic polyphenolic components, classified as neolignans (magnolol and honokiol) [2] or lignans (magnolin and fargesin derivatives) [3]. In the past two decades, the presence of other compounds belonging to the general group of plant phenolics, especially phenylethanoids [4,5], flavonoids [6] or phenolic acids [7], has also been confirmed in various herbal substances (organs) obtained from magnolia species. All of the aforementioned compounds are characterised by multidirectional biological potential, based mainly on antiradical and antioxidant properties [8]. They determine antibiodegenerative properties, confirmed by numerous scientific reports, which indicate the protective effect of magnolia extracts or polyphenolic isolates on the function of the nervous [9,10] and cardiovascular [11,12] systems, as well as their broad anti-inflammatory [13] and even anticancer [14,15,16] activities.

In order to conduct reliable studies of the biological activity of extracts and isolates from various plant species, it is necessary to use efficient methods for isolating complexes and/or individual bioactive components from plant material. A popular preparative procedure that is used by phytochemists to obtain herbal metabolites is liquid–solid extraction. In this case, the choice of extraction method and optimization of extraction conditions are critical parameters affecting the quality of the resulting product. For the isolation of polyphenolic compounds, ultrasound-assisted extraction (UAE) is often used [17,18,19,20]. When planning UAE, it is important to consider the influence of such parameters as solvent polarity, process time and temperature, ultrasound frequency, and the solvent-to-raw-herbal-material ratio [21]. One of new approach to strengthen the efficiency of various extraction processes is to use a computer-aided tool known as response surface methodology (RSM). This technique is based on optimising the responses (dependent variables) when two or more quantitative factors (independent variables) are involved. As a result, RSM minimises the number of runs needed and provides prediction of optimal extraction conditions and evaluation of other parameters (e.g., antioxidant properties) related to the optimised isolation of bioactive plant components [22]. To date, only one article has been published on the isolation of flavonoids from *M. offcinalis* leaves using infrared assisted extraction optimised with RSM [23]. Therefore, we decided to use a central composite design (CCD), which involved conducting several UAE experiments followed by RSM to determine the optimal settings for each factor used in the extraction. This research was also prompted by the fact that no phytochemical investigation of the polyphenolic composition and content in flower extracts of *M. soulangeana* var. ‘Lennei’ has been carried out so far. As this magnolia cultivar is a popular ornamental tree in Poland and its flower buds are easily available in spring in larger quantities for preparative purposes, we planned to assess the phytochemical and biological value of extracts obtained under optimised UAE conditions, assuming (in case of positive results) the possibility of scaling up the isolation process of biologically active components. Therefore, in addition to optimising UAE with RSM, an important objective of our work was to use advanced analytical tools, including coupled chromatographic, spectroscopic, and tandem mass spectrometric (ESI-QToF-MS/MS) techniques, to assess, in detail, the qualitative and quantitative profiles of polyphenolic compounds in MSL flower buds. Thus, we intended to fill the gap in the existing phytochemical and chemotaxonomic knowledge concerning the genus *Magnolia* L. The original aim of our study was also the evaluation, using RSM, of the antioxidant/antiradical potential of the total phenolic compounds present in MSL, as these are used to determine the broad antibiodegenerative activity described above for other magnolia species and may form the basis for further applications of this little-known herbal substance in therapy and/or dietetics.

## 2. Results

### 2.1. Multivariate Response Surface Modelling of Plant Material Extraction in Relation to Total Phenolic Content and Antioxidant Activity of Magnolia Extracts

Multivariate optimization methods used in RSM have currently been used extensively in both plant and food analysis. Recently, the most popular are central composite, Box–Behnken, and three-level factorial designs [24]. Table 1 shows the results of our experiments based on the CCD. Within the range of the established UAE parameters, the extraction time (4.77–55.23 min), ethanol concentration (16.36–83.64%), and solvent-to-raw-material ratio (13.18–46.82 mL/g) were analysed. In relation to the aforementioned conditions, different total phenolic contents (TPCs) and DPPH^•^ scavenging activities were observed for the individual MSL extracts obtained.

Using optimised UAE parameters, we determined that the TPC content ranged from 35.42 to 65.73 mg GAE/g of the dried herbal substance, and the percent of DPPH^•^ inhibition ranged from 86.76 to 94.29% under experimental conditions (Table 1). Overall, the highest TPC content was obtained for the following extraction parameters: time of 45 min, ethanol concentration of 70% (*V*/*V*), and solvent/raw-plant-material ratio of 40 mL/g. On the other hand, the highest scavenging activity (%I) was documented for the extract obtained during a 30 min extraction with 50% (*V*/*V*) ethanol, using a solvent-to-raw-material ratio of 30 mL/g. For both dependent parameters, the lowest TPC and %I were determined for extracts obtained during a 30 min extraction using a low (16.36%, *V*/*V*) ethanol concentration and 30 mL/g as the solvent/dried-herbal-material ratio. Therefore, the abovementioned results, obtained in the CCD multivariate modelling, suggested the existence of a potential correlation between ethanol concentration and the measured dependent parameters.

Regression coefficients for the TPC and %I models are presented in Table 2. For the TPC, the ethanol concentration (linear effect, *β*_2_; and quadratic effect, *β*_22_) and the interaction between the solvent concentration and solvent/raw-material ratio (coefficient, *β_23_*) were statistically significant (*p* < 0.05). For the antioxidant activity (%I model), the parameter intercept (coefficient *β*_0_) was statistically significant; however, the independent parameters had no significant effect on the percentage of DPPH^•^ inhibition.

The response surface model used for statistical evaluation showed a good fit for the independent variables, as evidenced by the high coefficient of determination and non-significance (*p* > 0.05) of the lack of fit parameter. The results of the ANOVA (shown in Table 3) revealed high coefficients of determination, namely *R*^2^ = 0.9685 and *R*^2^ = 0.9585, for TPC and %I, respectively, that confirmed a good correlation between responses and independent variables.

The analysis of TPC variance documented that all optimised parameters show statistical significance for the linear effect; however, for the quadratic effects, only the ethanol concentration was statistically significant (*p* < 0.05). When analysing interactions between individual parameters, the relationship between the ethanol concentration and solvent/raw-material ratio was seen to be statistically significant (*p* < 0.05). In regard to the antiradical activity (%I), the results obtained were negligible at *p* < 0.05. However, we simultaneously obtained a high value for the *R*^2^ coefficient in this analysis; therefore, despite a statistically insignificant match, we continued the study to identify the optimal UAE conditions related to %I as the dependent variable in the RSM model.

Based on the estimated parameters of the correlation coefficients, it is possible to establish an empirical relationship between the determined parameters, that is, the dependent (Y) and independent (X) variables [25]. Using the data shown in Table 2, these relationships are described by the following second-order polynomial Equations (1) and (2): (1)$$Y_{\mathrm{TPC}}\;(\mathrm{mg}\;\mathrm{GA}/g\;\mathrm{dry}\;\mathrm{wt}.)=24.60-0.53X_{1}+1.21X_{2}+0.29X_{3}+0.001X_{1}^{2}-0.012X_{2}^{2}-0.014X_{3}^{2}+0.001X_{1}X_{2}+0.01X_{1}X_{3}+0.01X_{2}X_{3}$$
(2)$$Y_{\%I}\;(\%)=58.95+0.28X_{1}+0.45X_{2}+1.19X_{3}-0.004X_{1}^{2}-0.004X_{2}^{2}-0.02X_{3}^{2}-0.0005X_{1}X_{2}-0.0005X_{1}X_{3}-0.0003X_{2}X_{3}$$
where Y_TPC_ represents total phenolic content; Y_%I_ is the antioxidant activity of MSL flower bud extracts; and X_1_, X_2_, and X_3_ are time (min), ethanol concentration (%), and solvent/dried-herbal-material ratio (mL/g), respectively.

In addition, the model fit for TPC was also demonstrated by correlating measured experimental versus predicted values and evaluating the random distribution of residuals (Figure 2a,b).

Finally, the overall RSM model showing the interactions between the independent factors used and the TPC/%I values is presented as three-dimensional response surface plots in Figure 3a–c and Figure 4a–c, respectively.

As can be seen in Figure 3a, as the concentration of ethanol in the extraction solvent increased, the TPC of the MSL extracts also rose. In this case, the extraction time had less of an effect on the TPC; however, the highest phenolic yield was observed for *t* > 50 min (optimum 55.2 min). Referring to Figure 2b, at a low solvent-to-raw-material ratio (<35 mL/g), the TPC decreased with the increasing time, but once this value was exceeded, the total amount of polyphenolic compounds in MSL extracts increased. Hence, we ultimately concluded that as the ratio and ethanol concentration increase, the TPC values of extracts obtained tend to increase. The highest concentrations of phenolics in MSL preparations were obtained for a solvent/raw-material ratio > 45 mL/g (optimum 46.8 mL/g), together with an ethanol concentration in the range of 63–82% (optimum 66.8%). As shown in Figure 3c, experimental UAE under optimal extraction parameters showed a higher mean content than the predicted TPC value, namely 76.73 vs. 66.55 mg GAE/g dry weight, respectively. We think that the discrepancies observed between the predicted and experimental values may be due to the influence of the temperature factor, including a possible increase in the temperature of the extraction medium by the generated high-energy ultrasonic waves. In all experiments, we maintained a constant extraction temperature (75 °C), additionally controlled by an external thermometer before starting each UAE process; however, especially with longer extraction times, this could have affected the increase in the experimental TPC value.

The literature data indicate that, besides the temperature, the extraction parameter that mainly affects the TPC content is ethanol concentration. Tabaraki and Nateghi [26] obtained the highest TPC content (6.05 and 6.21 mg GAE/g dry weight for the predicted and experimental values, respectively) by extracting rice bran with 67% ethanol at 54 °C for 40 min. The optimal high ethanol concentration (59%, *V*/*V*) in the RSM-monitored experiment was also estimated for the maximum phenolic yield (the predicted value of 157.35 mg GAE/g has been documented versus the experimental result of 149.12 mg GAE/g, calculated on a dry-weight basis) during a 25 min UAE of pomegranate peel at 80 °C, using a plant-material-to-solvent ratio of 1:44 [27].

With regard to our study using the DPPH^•^ assay, as shown in Figure 4a–c, the optimal extraction parameters for the highest antioxidant/antiradical activity (%I) of MSL flower bud extracts were an extraction time of 28.1 min, 59.6% ethanol as extraction solvent, and a solvent-to-plant-material ratio of 29.6 mL/g. The antioxidant activity of extracts obtained under optimal conditions was 87.10%, was lower than the predicted value (94.04%). In our opinion, the discrepancies in the abovementioned results may be due to the fact that RSM, as a statistical tool, theoretically assumes an optimal model for the diffusion of phenolic components from the plant matrix into the extraction medium based on the selected independent variables (X_1_, X_2_, and X_3_). However, when comparing the UAE parameters, established in our two experiments on the antioxidant activity of MSL flower bud extracts, it is readily apparent that the optimal extraction time in the I% model (using the DPPH^•^ assay) was almost twice as short (28.1 vs. 55.2 min), and the solvent-to-solid ratio was about two-thirds of that used in the TPC model (29.6 vs. 46.8 mL/g). In addition, the optimum ethanol concentration (59.6%), affecting the elution strength of the extractant, was lower in the former model compared to the latter procedure, which used 66.8% ethanol. As a result, less efficient diffusion and mass transfer of plant phenolics into the extraction medium in the I% model should be suspected under real experimental conditions. In our opinion, this was also experimentally confirmed by the slightly lower antioxidant activity (I%) values compared to those predicted.

A study on RSM-controlled optimisation of extraction parameters to obtain the highest antioxidant activity of plant extracts, as measured by the DPPH^•^ assay, was published by Wijngaard and Brunton [28]. In contrast to our experiments, the researchers used a simple vortex-assisted solid–liquid extraction procedure and included the temperature as one of the independent variables in the response surface design. They determined the extraction conditions (31 min, 80 °C, and 56% ethanol as extraction solvent) for the optimal antioxidant activity of apple pomace extracts. The quantitative results obtained with the abovementioned extraction conditions and calculated as Trolox equivalents (TEs) were 403 and 449 mg TE/100 g dry weight in relation to the predicted and experimental values, respectively.

### 2.2. Phytochemical Qualitative Profiling of MSL Flower Bud Components Using Coupled Chromatographic, Spectroscopic, and Tandem Mass Spectrometric Techniques

A total of 22 polyphenolic antioxidants (Figure 5 and Table 4) were identified in MSL flower bud extracts that were prepared in triplicate under RSM-optimised UAE conditions, namely using 66.8% (*V*/*V*) ethanol as the extraction solvent, an extraction time of 55.2 min, and a ratio of extractant volume to dried herbal substance of 46.8 mL/g.

For a qualitative profiling, RP-LC/PDA and LC/PDA/ESI-QToF/MS-MS methods were employed simultaneously. The former method was very important for obtaining characteristic UV spectra (of high quality) for the examined compounds and an additional set of retention time parameters, collected under conditions of better peak resolution during optimised classical RP-LC/PDA chromatographic analysis, which could be further compared with data obtained for reference polyphenolic components (Table 4). In the next step, compound identification was performed based on a data set related to specific precursor ions and fragmentation patterns that were acquired using Agilent MassHunter Workstation Qualitative Analysis 10.0 Software. It was found that the negative-ionization mode (NEG) was more suitable, especially for obtaining deprotonated polyphenolic molecules and efficient cleavage of precursor ions that facilitated their proper identification. A high-resolution tandem mass spectrometer (Q-ToF) effectively delivered precursor (molecular) ions by operating in scan mode (MS1), while MS2 mode allowed for the selection of product (fragment) ions with a specific *m*/*z* value, formed by ion dissociation in the collision chamber. Deprotonated ions [M − H]^−^ and main product ions reported in the MS spectra of all phenolics are shown in Table 4. MS data acquired for MSL components were also compared with the spectral data obtained for reference polyphenolic substances.

Detailed results of qualitative profiling of MSL flower bud components using coupled chromatographic, spectroscopic, and tandem mass spectrometric techniques are discussed below, and the molecular structures of the main polyphenolic antioxidants are shown in Figure 6.

#### 2.2.1. Phenolic Acids

Hydrophilic constituents, recorded at low retention times (from ~3.00 to 13 min), were analysed first. Compound **1** (R_t_ ~ 3.67 min) was fragmented to a precursor ion [M − H]^−^ with *m*/*z* 153.0193, and to a product ion [M − H − CO_2_]^−^ with *m*/*z* 109.0297 which was formed from the loss of CO_2_ from the carboxylic acid. MS data published by other researchers [29,30] have unequivocally confirmed that the molecule sought is one of the hydroxybenzoic acids (protocatechuic acid). Compound **2**, with an average R_t_ of 4.58 min, was identified through the analysis of the precursor ion [M − H]^−^, with *m*/*z* 353.0855, that released a fragment ion of *m*/*z* 191.0543 (quinic acid). The chromatographic and spectrometric data also corresponded with those previously reported by Saravanakumar et al. [30] and Park et al. [31], matching the fragmentation pattern of chlorogenic acid (C_16_H_18_O_9_), which belongs to the group of hydroxycinnamic acids. The identity of compound **5** (with R_t_ ~ 12.41 min), giving precursor ion [M − H]^−^ with *m*/*z* 167.0351 and characteristic cleavage to product ions (with *m*/*z* 135.0116 and *m*/*z* 108.0210), was confirmed with a certified reference substance and published data [31,32] as vanillic acid (C_8_H_8_O_4_).

#### 2.2.2. Phenylethanoids

The second group of polyphenolic antioxidants identified in MSL flower bud extracts were phenylethanoids, for which specific molecular-structure elements were confirmed, namely caffeoyl and phenylethanol moieties, along with the presence of rhamnose and glucose.

Compound **3**, with R_t_ ~ 5.08 min and a characteristic UV spectrum showing two maxima (198 and 330 nm), provided a [M − H]^−^ precursor ion of *m*/*z* 785.2466, further yielding a fragment ion (*m*/*z* 623.2127), corresponding to the loss of hexose. Other characteristic products were also obtained, including an ion with *m*/*z* 477.1619 and 315.0898 (formed from the previous one by the cleavage of glucose = 162 Da), as well as a product ion of *m*/*z* 179.0338 (C_9_H_7_O_4_), corresponding to a caffeoyl moiety. All of these product ions were further confirmed by a fragmentation pattern of the reference substance and data from the literature [33,34]. The compound was identified as echinacoside (C_35_H_46_O_20_). Compound **4** yielded a characteristic precursor ion with *m*/*z* 931.3058 and fragment ions with *m*/*z* 769.2726, 751.2572, 179.0499, and 161.0238 corresponding to the loss of rhamnose, glucose, and caffeoyl moiety (similarly to compound **3**) and, in the next step, the loss of rhamnose. The fragmentation pattern was also similar to that described in the literature [33,34]; therefore, the compound was tentatively identified as 2′-rhamnoechinacoside (C_41_H_56_O_24_). As for compound **9** (R_t_ ~ 26.48 min), a [M − H]^−^ precursor ion with *m*/*z* 623.1985 was formed, and characteristic product ions with *m*/*z* 461.1665 (resulting from the loss of the caffeoyl moiety) and 179.0342 were also found. When the H_2_O molecule was detached, a fragment ion with *m*/*z* 161.0246 was formed. Based on the LC/MS analysis of product ions obtained for the reference substance and information published by Joo et al. [35] for *M. denudata* flower extracts, compound **9** was identified as acteoside (verbascoside).

#### 2.2.3. Flavonoids

As a medium-polar group of polyphenolic antioxidants, various types of flavonoid compounds were found in MSL flower bud extracts. Compound **7** was identified on the basis of a precursor ion [M − H]^−^ with *m*/*z* 609.1433 and a predominant aglycone ion with *m*/*z* 301.032, which indicated the loss of rhamnose and glucose moiety from the molecular structure of flavonoid glycoside. Additionally, [M − H − CO − H]^−^ and [M − H − CO_2_ − H]^−^ ions were formed, with *m*/*z* of 271.0253 (C_14_H_9_O_6_) and 257.0411 (C_14_H_9_O_5_), as a result of the loss of CO and CO_2_, respectively. They were characteristic for the *retro*-Diels–Alder (*r*DA) fragmentation pattern for quercetin as a flavonol aglycone. The literature data [36,37] referring to studies of other magnolia species further helped to identify compound **7** as quercetin 3-*O*-rutinoside (rutoside). Compound **6**, which preceded compound **7** on the base peak chromatogram (R_t_ ~ 19.24 min), formed a precursor ion of *m*/*z* 609.1303, similar to the molecular ion of compound **7**. According to a study published by Abad-Garcia et al. [38], distinguishing between these compounds is possible based on the analysis of fragment ions formed by the loss of rhamnose and rhamnosyl-glucose (rutinose) and the intensity of these ions. This made it possible to differentiate the two flavonol components and identify compound **6** as quercetin 3-*O*-neohesperidoside (C_27_H_30_O_16_). A molecular ion of *m*/*z* 463.0889 was formed for compound **8** (R_t_ ~ 24.37), and further MS/MS fragmentation, after detaching the glucose molecule, yielded a fragment ion with *m*/*z* 301.0347, analogous to the one previously identified in the rutoside fragmentation pattern. UV and retention data (Table 4) compared with the reference substance, as well as information provided by other researchers [36], who analysed magnolia extracts, allowed us to identify compound **8** as quercetin 3-*O*-glucopyranoside (C_21_H_20_O_12_), known by its usual name, isoquercitrin.

In the case of compound **10** (with R_t_ ~ 28.89 min), a precursor [M − H]^−^ ion of *m*/*z* 593.1571 was obtained which, upon cleavage, released the [MH-146-162]^−^ ion with *m*/*z* 285.0444, characteristic of the kaempferol aglycone, with the observed loss of the rhamnosyl moiety. Taking into account the UV, retention, and other published MS data [36,39], compound **10** was identified as kaempferol 3-*O*-rutinoside (nicotiflorin).

A spectrometric analysis of compound **12** (R_t_ ~ 30.80 min) confirmed the presence of a third group (in addition to the quercetin and kaempferol derivatives described above) of flavonoid glycosides in MSL flower buds. This phenolic constituent released a molecular ion with *m*/*z* 477.1042. In the course of the MS/MS analysis, the loss of glucose (162 Da) was documented, and characteristic ions with *m*/*z* 315.0486 (corresponding to isorhamnetin moiety) and *m*/*z* 271.0256 were formed. Similar results were published by Sokkar et al. [36] and confirmed with the MS fragmentation pattern obtained for the reference substance. Thus, compound **12** was identified as isorhamnetin 3-*O*-glucoside (C_22_H_22_O_12_). Compound **11** yielded a precursor [M − H]^−^ ion with *m*/*z* 623.1622 and fragment ions with *m*/*z* 315.0516 and *m*/*z* 271.0252, corresponding to isorhamnetin aglycone molecule, and a loss of 308 Da, indicating the presence of rhamnosyl-glucose (rutinose) attached to the aglycone, respectively. Therefore, compound **11** was identified as isorhamnetin 3-*O*-rutinoside (narcissin, C_28_H_32_O_16_). Compound **13** (R_t_ ~ 35.50 min) had a similar UV spectrum to compound **11** and the same molecular mass; hence, it was suspected to be an isorhamnetin derivative. It released a precursor ion with *m*/*z* 623.1611 and a fragment ion with m/z 315.0509, which was very similar to compound **13**, so this component was described as the 3-*O*-rutinoside isomer of isorhamnetin.

Compound **14** released a molecular [M − H]^−^ ion at *m*/*z* 637.1728 and a product ion at *m*/*z* 330.0684 that further yielded a methyl unit, giving another product ion at *m*/*z* 315.0509 that corresponded to the isorhamnetin aglycone. Fragmentation results were compared with the literature [40], and compound **14** was finally identified as rhamnazin 3-*O*-rutinoside (ombuoside, C_29_H_34_O_16_).

#### 2.2.4. Lignans

The group of hydrophobic phenolic constituents of MSL that was identified in MSL extracts was composed of lignans. These compounds were recorded at retention times above 56 min (Figure 5). The taxonomic origin of *M. soulangeana* (as a hybrid of *M. liliiflora* and *M. denudata*) prompted us to look for a group of phenolic compounds previously described for these two taxa [7,31,35]. However, due to the rapid breakdown of lignans, even with low collision energy used, it was difficult to accurately determine the chemical structure of these compounds. Therefore, in the qualitative profiling of MSL extracts, we relied mainly on the characteristic UV and MS spectra of reference substances and the identification of unique product ions derived from lignan structures. Compound **15** (R_t_ ~ 56.22 min) released a precursor ion with *m*/*z* 415.4612 and main product ions with *m*/*z*: 221.1545 and 236.1059. In addition, we documented three maxima (204, 230, and 278 nm) in its UV spectrum. Retention, spectroscopic, and MS/MS data were compared with the reference substances and confirmed with the results of other researchers [41,42]. Finally, compound **15** was tentatively identified as magnolin (C_23_H_28_O_7_). Compound **18** (R_t_ ~ 62.25 min) yielded ions with *m*/*z* 357.1360 and 242.9433, as well as characteristic product ions with *m*/*z* 174.9563 and 112.9856, also reported for other suspected lignan compounds. The UV maxima (202, 234 and 284 nm) and MS spectra of this component corresponded to the spectra of the reference substance and MS data published [42,43]. Therefore, compound **18** was described in Table 4 as fargesin (C_21_H_22_O_6_).

In regard to compounds **16**–**17** and **19**–**22,** their UV spectra indicated the lignan structure resembling fargesin, and they yield the same characteristic product ions with *m*/*z* 174.956 and 112.9856. After comparing the results of our MS/MS analysis with the published data on *Magnolia* species [43], we decided to include all of these compounds in the group of furofuran lignans, with the molecular structure type of fargesin.

An interesting result of our qualitative study was the absence of neolignans (including honokiol and magnolol and their derivatives) in the MSL extracts examined. This confirmed the similarity of the lignan profile to *M. liliiflora* and *M. denudata,* as well as to an important medicinal taxon, *M. biondii*, whose monograph (*Magnoliae biondii flos*) is listed in the European Pharmacopoeia [42,43,44].

### 2.3. Phytochemical Quantitative Profiling of Polyphenolic Antioxidants in MSL Extracts Using RP-LC with a Photodiode Array (PDA) Detection

In terms of the detailed quantitative analysis of polyphenolic antioxidants, MSL flower bud extracts were prepared under RSM-optimised UAE conditions, similarly to the qualitative profiling procedure. On the basis of the studies described in Section 2.2, we selected four major groups of compounds (including flavonoids, phenylethanoids, phenolic acids, and lignans), whose contents were calculated for the leading components, using the corresponding reference substances. The main results of the research that was performed is shown in Figure 7.

As can be seen, phenylethanoids and flavonoids (quercetin derivatives) were the most abundant group of polyphenolic antioxidants in MSL flower buds. Their average content was established at ~30.3 and ~30.5 mg/g of the dried herbal substance, respectively (Figure 7a), which corresponded to more than 40% of the total phenolic antioxidant content determined in the plant material examined (Figure 7b). In the group of quercetin derivatives (Q-der), rutoside accounted for about 95% of the quantified constituents, demonstrating the very high content of this flavonol in MS flower buds. The presence of rutoside, as the dominant component in flowers of other magnolia taxa, and its antioxidant and anti-inflammatory potential were also confirmed by other researchers [13,31,36]. Other quantified flavonoid compounds (kaempferol and isorhamnetin derivatives) may have a much weaker effect on the antioxidant/antiradical potential of MSL flower bud extracts, as their percentages of total phenolic content were about 4.7 and 3.6%, respectively (Figure 7b). The second important group of polyphenols in MSL flowers which can significantly affect the antioxidant and antiradical capacity of the aqueous–ethanolic extracts obtained thereof are phenylethanoid compounds (Figure 7a). In this group, the dominant polyphenolic components in the herbal substance examined were 2′-rhamnoechinacoside and acteoside (verbascoside), whose average content reached 19.2 and 8.6 mg/g dry wt., respectively. The results of phytochemical and biological studies that were conducted for other magnolia species [4,5,33] highlight the significant protective effects of phenylethanoids against free-radical-induced oxidative damage. When considering the antioxidant/antiradical activity of MSL lignan compounds—mainly fargesin derivatives—it is important to emphasize the increasing number of phytochemical and biological studies of these constituents in the genus *Magnolia* L., showing their antibiodegenerative (anti-inflammatory, antimicrobial, and antineoplastic) potential [3,41,42,43]. In view of these studies, we are hopeful that a relatively high content of these compounds, accounting for more than 5% of the total polyphenolic components, was determined in MSL flowers.

## 3. Materials and Methods

### 3.1. Plant Material and Its Pre-Treatment

Unopened flower buds of *M. soulangeana* var. ‘Lennei’ (MSL) were collected in early spring from the specimens growing in the arboretum of Maria Curie-Sklodowska University (UMCS) Botanical Garden (geographical coordinates: 51°16′ N; 22°30′ E; Lublin, Poland) in the presence of a botanical taxonomy specialist employed by the garden. The botanical identification of the plant material was also confirmed by a certificate (No. 119/2023) issued by Dr. Agnieszka Dąbrowska (senior specialist), representing the UMCS Botanical Garden. After collection, the fresh plant material was dried immediately in a laminar ventilated dryer at a temperature not exceeding 35 °C, and a small portion of MSL buds (a voucher specimen) was then deposited in the Department of Pharmacognosy at the Medical University of Lublin. The remaining plant material was ground into a fine powder, using a laboratory mill, and sieved to obtain particles of 0.75 mm. Approximately 50.0 g of the powdered herbal substance was then placed in a sealed vessel and used in further UAE experiments. Before starting the extraction procedures, the average moisture content was determined for 1.000 g samples of powdered MSL buds by drying them in an oven at 105 °C to a constant weight.

### 3.2. Solvents, Reagents, and Certified Reference Substances

Ethanol 99.8% (EtOH) and methanol (MeOH) provided by Avantor Performance Materials (Gliwice, Poland) were of analytical grade. Other solvents (acetonitrile and formic acid) used in the qualitative and quantitative analysis of MSL phenolics were of chromatographic or LC/MS grade and were purchased from J. T. Baker (Gross-Gerau, Germany). Ultrapure water was obtained from a Direct-Q system (Millipore, Molsheim, France). The reagents used for the antioxidant spectrophotometric assays, namely FCR (Folin–Ciocâlteu reagent), calcium carbonate, gallic acid, and DPPH^•^ (2,2-diphenyl-1-picrylhydrazyl radical), were supplied by Sigma-Aldrich Chemie GmbH (Steinheim, Germany). Certified reference substances of plant phenolics, listed below, had a purity higher than 95%. Acteoside (verbascoside), rutoside, fargesin, and magnolin were purchased from PhytoLab GmbH (Vestenbergsgreuth, Germany). Honokiol and magnolol were obtained from ChromaDex Inc. (Santa Ana, CA, USA). Phenolic acids (protocatechuic, vanillic, and chlorogenic), flavonoids (quercitrin, isoquercitrin, nicotiflorin, and isorhamnetin 3-*O*-glucoside), and echinacoside were supplied by Sigma-Aldrich Chemie GmbH (Steinheim, Germany). Stock solutions of each reference substance were prepared in methanol (LC-grade) in the concentration range of 0.1–0.2 mg/mL and stored at 4–8 °C. Prior to chromatographic analysis, appropriate standard dilutions with methanol were prepared from each stock solution, and calibration curves with at least five levels were determined.

### 3.3. Central Composite Design and Response Surface Methodology

To optimize the process of ultrasound-assisted extraction (UAE), central composite design (CCD) modelling was used as an efficient variant used in response surface methodology [24]. The optimization procedure consisted of 15 different experimental sets containing 8 factorial points, 6 axial points, and 2 replicates of central points. Time of extraction, X_1_ (min); solvent concentration, X_2_ (%); and solvent/plant-material ratio, X_3_ (mL/g), were chosen as independent variables (Table 5). The selection of these variables for CCD and the basic range for each parameter, including extraction time, X_1_ (from 15 to 45 min); ethanol concentration, X_2_ (from 30 to 70%); and solvent/plant-material ratio, X_3_ (from 20 to 40 mL/g), was based on preliminary single-factor experiments with MSL flower buds, which revealed satisfying yields (response values) of the individual phenolics determined via the RP-LC/PDA method. A three-factor response surface optimisation was then initiated using the experimental parameters shown in Table 5. We used the module of the Design and Analysis of Central Composite Experiment (Statistica 13.3.0) to extend the range of independent variables and to establish the axial points (upper and lower limits for X_1_, X_2_, and X_3_) for the orthogonal CCD.

The total phenolic content (TPC) and antioxidant activity (%I) were chosen as dependent variables in CCD. To evaluate these variables in MSL extracts, the DPPH^•^ radical assay and a method based on the Folin–Ciocâlteu reagent reaction were selected, as they are analytical procedures that are commonly used in phytochemical studies and allow for a comparison of the results obtained by different researchers.

The multivariate data obtained in the CCD were further fitted to a second-order (quadratic) polynomial model of RSM [25], using the equation shown below:Y_i_ = *β*_0_ + *β*_1_X_1_ + *β*_2_X_2_ + *β*_3_X_3_ + *β*_11_X_1_^2^ + *β*_22_X_2_^2^ + *β*_33_X_3_^2^ + *β*_12_X_1_X_2_ + *β*_13_X_1_X_3_ + *β*_23_X_2_X_3_
(3)
where Y_i_ is the predicted response; *β*_0_ is the intercept; *β*_1_–*β*_3_, *β*_22_ and *β*_33_, and *β*_13_ and *β*_23_ stand for linear, quadratic, and cross product regression coefficients, respectively; and X_1_, X_2_, and X_3_ are independent variables.

### 3.4. Ultrasound-Assisted Extraction (UAE) and the Preparation of Extracts for Antioxidant and Phytochemical Studies

The process of obtaining extracts from dried, powdered MSL buds (1.000 g), placed in round-bottom glass flasks under a reflux condenser, was performed in a Sonorex RK 255H ultrasonic bath (Bandelin, Berlin, Germany) according to the detailed extraction conditions established in the central composite modelling described in Section 3.3. For this purpose, different concentrations of EtOH (16.36–83.64%) were prepared, and various volume ratios of solvent to herbal substance (13.18–46.82 mL/g) and extraction times (4.77–55.23 min) were used according to the characteristics of independent variables (Table 5) determined by the CCD. The physical parameters that were held constant throughout the UAE procedure were the ultrasound frequency (35 kHz) and power density (~16.5 W/L) of the ultrasonic device. All experiments started when the water-bath temperature in the UAE apparatus reached the set value of 75 °C. After the extraction was completed and the herbal preparations were cooled, each extract was filtered through Filtrak paper (No. 388) into a receiver. The remaining plant material was washed twice with 20 mL of the solvent used for the main UAE process and filtered. The combined ethanol–water extracts were then evaporated to dryness under vacuum, and each dry residue was dissolved in several portions of 75% (*V*/*V*) MeOH and transferred to a calibrated flask (25 mL), which was then refilled to nominal volume with the same solvent. In this way, crude primary extracts from MSL flower buds were obtained for antioxidant-activity studies. To remove co-extractable ballast substances (chlorophyll) from the MSL extract obtained under optimised (by RSM) extraction conditions and to obtain a pure phenolic fraction for the qualitative and quantitative analysis, a validated solid-phase extraction (SPE) procedure on BakerBond octadecyl columns (500 mg sorbent weight, 3 mL), developed by Zgórka [45], was used. The SPE eluates, obtained in triplicate, were then subjected to phytochemical profiling, as described in Section 3.7 and Section 3.8.

### 3.5. Total Phenolic Content Assay

The total phenolic content (TPC) was determined in MSL flower extracts according to the spectrophotometric method developed by Zgórka et al. [46] (with slight modifications), using a Folin–Ciocâlteu reagent (FCR). Briefly, 1 mL of each primary MSL extract was pipetted into a 25 mL calibrated flask and diluted with distilled water. Then, 1 mL of the diluted extract, 5 mL of distilled water, and 0.5 mL of FCR were added to a volumetric flask (10 mL) and vortexed for 2 min. Afterwards, 1.5 mL of 20% aqueous sodium was added, and the calibrated vessel was made up to the nominal volume with distilled water, followed by vortexing for another 2 min. The prepared sample was allowed to stand in darkness for 1 h, and, after this time, the absorbance was measured using a 10S Series UV-Vis Spectrophotometer (Thermo Electron Scientific Instruments, Madison, WI, USA) at 765 nm, against a reference solution (1 mL of the diluted primary MSL extract and 9 mL of distilled water). Since the TPC was determined as the gallic acid equivalent (GAE), methanolic solutions of this reference substance (C = 0.05–0.25 mg/mL) were prepared, and the six-point calibration curve was constructed following the same aforementioned procedure. The spectrophotometric protocol was performed in triplicate for both the sample examined and the standard substance. Finally, the TPC was calculated as mg GAE per 1 g MSL flower buds (dry weight).

### 3.6. Antioxidant (Antiradical) Activity Assay

The radical scavenging capacity of different samples was measured using 2,2-diphenyl-1-picrylhydrazyl (DPPH^•^) radical. Antioxidant activity testing was performed by applying the method described by Benabdallah et al. [47], with slight modifications. Briefly, each sample was prepared at a concentration of 1 mg/mL with 50% methanol. To 20 μL of samples, 180 μL of methanolic solution of DPPH^•^ (0.2 mM) was then added. The 96-well plates with mixtures were incubated in the dark for 30 min at room temperature, and the absorbance was measured at a wavelength of 515 nm against the blank (methanol), using an ELx808 Absorbance Microplate Reader (BioTek, Winooski, VT, USA). The radical scavenging activity, which was expressed as percentage of inhibition (%I), was calculated using Formula (4):%I = (A_control_ − A_sample_/A_control_) × 100 (4)
where A_control_ is the absorbance of DPPH^•^ solution without extracts, and A_sample_ is the absorbance of the samples at 515 nm.

### 3.7. RP-LC/PDA Qualitative and Quantitative Analysis

The qualitative and quantitative analysis of MSL extracts was carried out using an Agilent Technologies Model 1100 liquid chromatograph (Waldbronn, Germany) equipped with a Rheodyne manual injector and photodiode array detector (PDA) set at 215 nm (lignans), 254 nm (flavonoids and hydroxybenzoic acids), and 325 nm (hydroxycinnamic acids and phenylethanoids). The chromatographic separation of polyphenolic compounds was performed on an Aquasil C18 stainless-steel column (250 × 4.6 mm I.D., dp = 5 μm). To obtain the sufficient separation of all components, the gradient elution program for was developed. A binary solvent system was used that consisted of 1 mM H_3_PO_4_ (A) and acetonitrile (B), at a flow rate of 1 mL/min, as follows: 0 min/15; 15 min/15; 25 min/20; 35 min/20; 55 min/45; and 60 min/95% B, continued isocratically to 65 min. The post time was set to 10 min. The injection volume was 10 μL. UV spectra of all phenolics were recorded within the range of 190–400 nm. The identification of individual compounds was performed by comparing retention times of the peaks obtained and their UV spectra with those of the reference substances. Spectral data acquisition was conducted using Agilent ChemStation Rev. A.10.02 software. In terms of the quantitative analysis of all MSL phenolics, an external standard method was used. For this purpose, five-point calibration curves were constructed using methanolic solutions (C = 0.01 to 0.20 mg/mL), which were prepared as the dilutions of the stock solutions of the certified reference substances. The linearity of calibration curves referring to individual compounds was assessed using regression coefficients (*R*^2^). Samples were analysed in triplicate.

### 3.8. Qualitative Profiling of MSL Phenolics Using RP-LC/PDA/ESI-QTOF/MS-MS Method

The qualitative analysis was performed using an Agilent Technologies system (Santa Clara, CA, USA) consisting of an LC 1290 Infinity chromatograph coupled to a PDA detector and a 6530B QTOF-MS/MS mass spectrometer and equipped with an electrospray ionization (ESI) source. Chromatographic separation of polyphenolic compounds was performed on a Zorbax Stable Bond-C18 narrow-bore column (2.1 × 150 mm, dp = 3.5 μm). Volumes of the injected sample aliquots were 10 μL. A mobile phase gradient (at a flow rate of 0.2 mL/min) composed of acetonitrile (B) and water (A) with 0.1% (*V*/*V*) formic acid was employed as follows: 0 min/15; 15 min/15; 55 min/55; and 65 min/95 and 72 min/95% B in A. The column re-conditioning time was 12 min. The mass spectra of compounds examined were recorded in the negative-ionization mode in the range of 100–1000 *m*/*z*, using Agilent MassHunter Workstation Qualitative Analysis 10.0 Software. The collision-induced dissociation (CID) energies were set to −20 and −40 eV to obtain MS/MS spectra with the highest intensity of product ions. The confirmation of the molecular structure for compounds examined was conducted on the basis of their fragmentation patterns compared with data recorded for certified reference substances and published in freely available MS databases.

### 3.9. Statistical Modelling

All statistical analyses used in RSM modelling (experimental design and regression analysis of the experimental data) were performed using Statistica 13.3.0 (TIBCO Software Inc., Palo Alto, CA, USA). Linear and quadratic effects and two-way interaction models were also selected for the statistical evaluation. Model adequacy was evaluated using the lack of fit, as well as the coefficient of determination (*R*^2^), as obtained from the analysis of variance (ANOVA). In addition, the correlations between the values observed vs. predicted values and graphically evaluated residuals vs. predicted values were assessed.

Statistical evaluation in the quantitative RP-LC/PDA profiling of individual polyphenolic compounds and antioxidant/antiradical assays was performed using the GraphPad Prism 5 programme (GraphPad Software, San Diego, CA, USA) with the F-Snedecor test in the one-way analysis of variance. Statistical significance was set at *p* < 0.05.

## 4. Conclusions

In this study, we present for the first time the results of the RSM-controlled experiments on the optimisation of ultrasound-assisted extraction of dried flower buds of *M. soulangeana* var. ‘Lennei’ that were performed in order to obtain both high yields of polyphenolic constituents and to determine their antioxidant potential. The key findings for optimal extraction conditions were as follows: 66.8% ethanol as extraction solvent, an extraction time of 55.2 min, and a solvent-to-solid ratio of 46.8 mL/g.

It is also the first report on advanced chromatographic, spectroscopic, and spectrometric analyses of the qualitative profile and content of polyphenolic compounds, providing new data on the chemical composition of this ornamental magnolia taxon. A simultaneous phytochemical and biological analysis confirmed that MSL flower buds could serve as a potential future source of bioactive polyphenolic antioxidants, with promising therapeutic (anti-inflammatory) effects, and, like some other magnolias, they even have potential dietary applications.

## Figures and Tables

**Figure 1 molecules-28-06335-f001:**
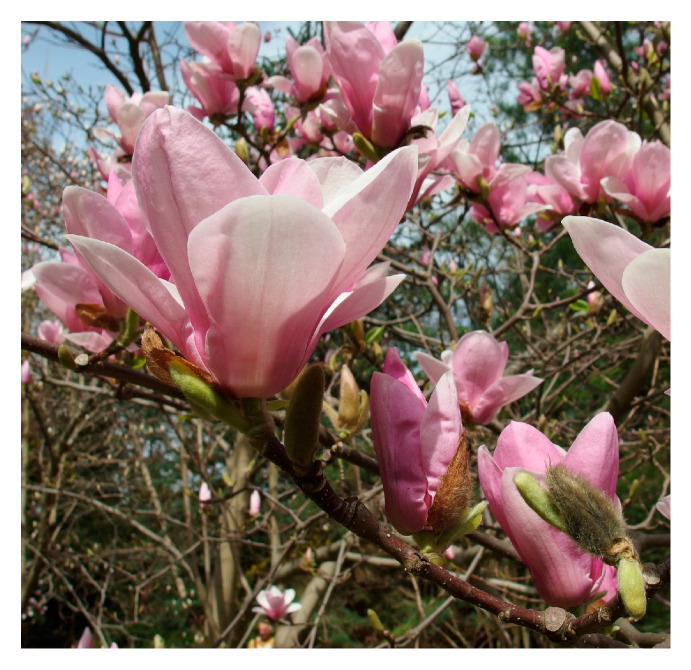
*M. soulangeana* Soul.-Bod. var. ‘Lennei’ during flowering (photo: A. Adamska-Szewczyk).

**Figure 2 molecules-28-06335-f002:**
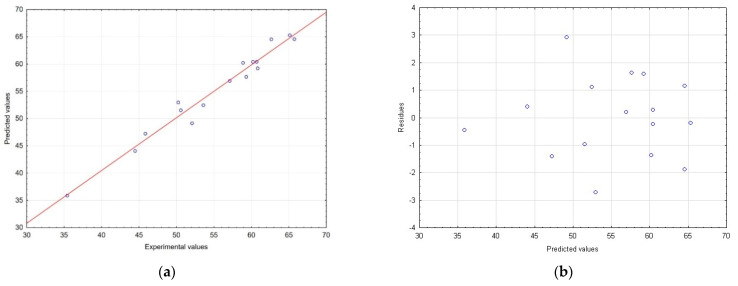
Correlation between the predicted and experimental values (**a**) and distribution of residuals vs. predicted values obtained (**b**) while determining TPC.

**Figure 3 molecules-28-06335-f003:**
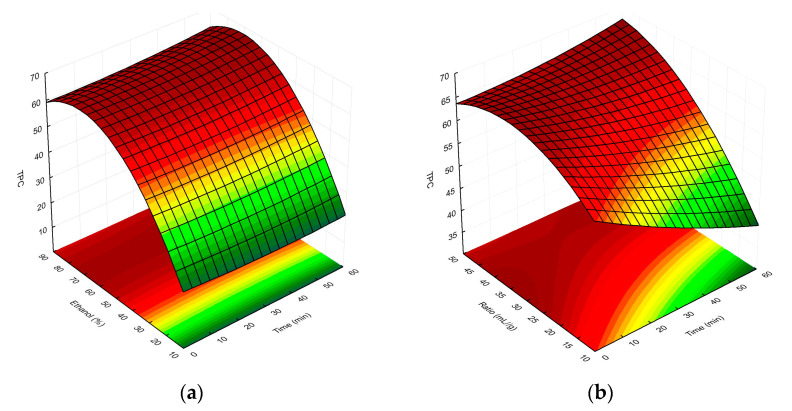
Surface plot showing the effect of ethanol concentration vs. extraction time, (**a**) extractant-volume-to-plant-material ratio vs. extraction time (**b**), and extractant-volume-to-plant-material ratio vs. ethanol concentration (**c**), respectively, on TPC in the MSL extracts. The red and green colours in the graph indicate the highest and lowest TPC values, respectively.

**Figure 4 molecules-28-06335-f004:**
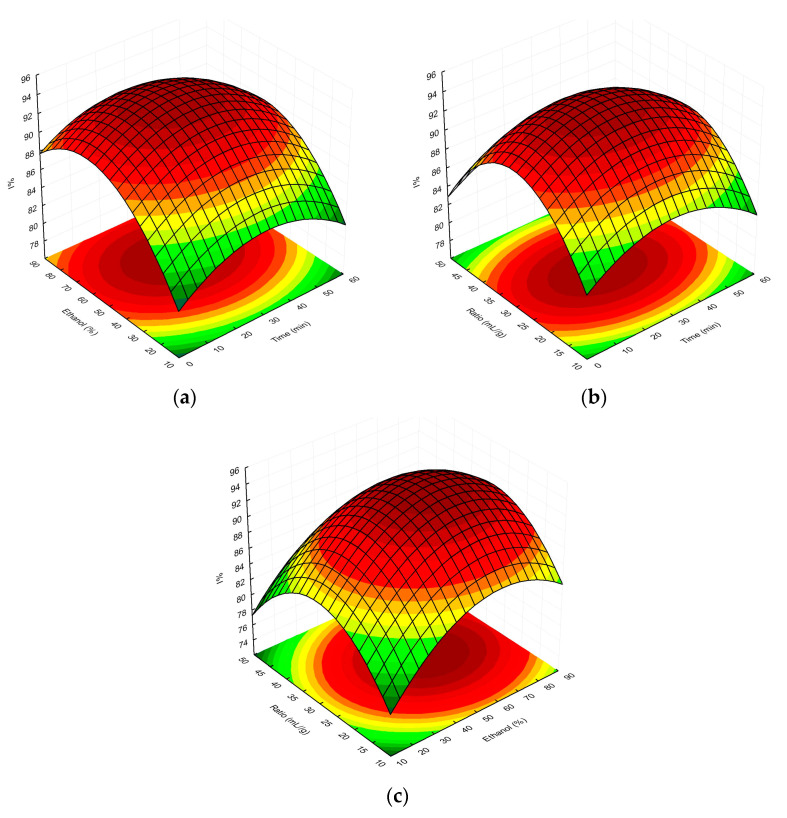
Surface plot showing the effect of ethanol concentration vs. extraction time (**a**), extractant-volume-to-plant-material ratio vs. extraction time (**b**), and extractant-volume-to-plant-material ratio vs. ethanol concentration (**c**), respectively, on %I determined for MSL extracts. The red and green colours in the graph indicate the highest and lowest %I values, respectively.

**Figure 5 molecules-28-06335-f005:**
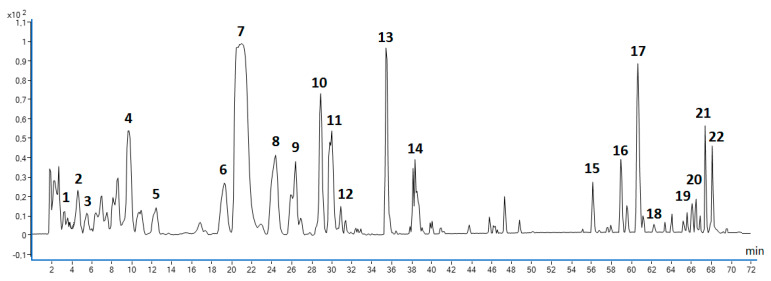
Base peak chromatogram (BPC) showing polyphenolic compounds identified in MSL flower buds using LC/ESI-QToF/MS-MS method in negative-ionisation mode; numbering of individual components according to Table 4.

**Figure 6 molecules-28-06335-f006:**
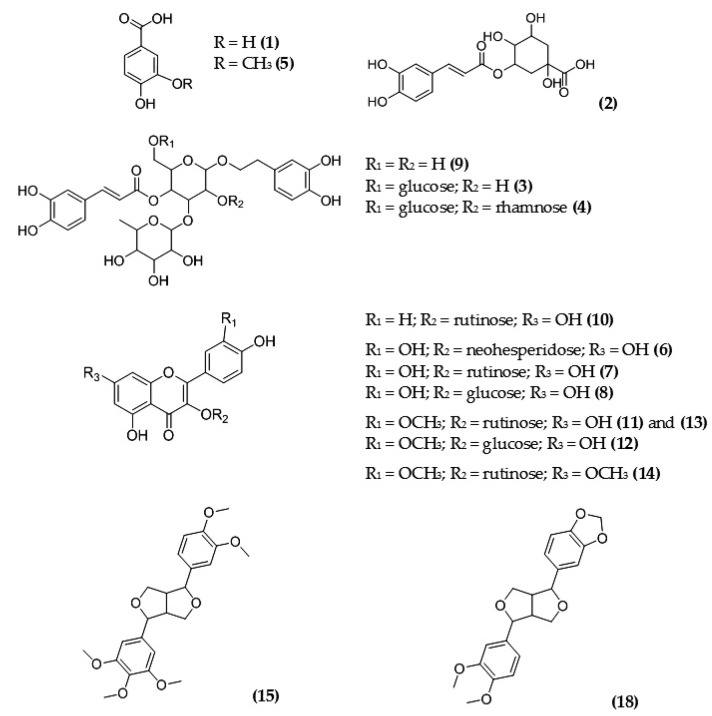
Chemical structures of the main polyphenolic compounds identified in MSL flower buds; numbering of individual components (in parentheses) is as presented in Table 4.

**Figure 7 molecules-28-06335-f007:**
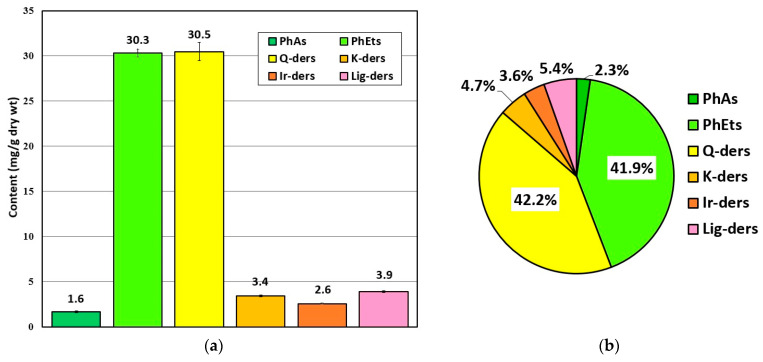
The mean content (mg/g dry wt.) of polyphenolic antioxidants in MSL flower buds (**a**) and the percentage ratio of individual components in total polyphenolic content quantified in this herbal substance (**b**). Explanations: PhAs—phenolic acids; PhEts—phenylethanoids; Q-ders—quercetin derivatives; K-ders—kaempferol derivatives; Ir-ders—isorhamnetin derivatives; Lig-ders—lignan derivatives.

**Table 1 molecules-28-06335-t001:** Run coded levels, total phenolic content (TPC), and DPPH^•^ scavenging activity (%I), as obtained under the experimental (Exp.) and predicted (Predict.) CCD conditions.

Run	Coded Levels			TPC (mg GAE/g dry wt.)		%I	
	X_1_	X_2_	X_3_	Exp. *	Predict.	Exp. *	Predict.
*Factorial points*							
1	−1(15)	−1(30)	−1(20)	50.58	51.55	88.25	88.01
2	−1(15)	−1(30)	1(40)	52.08	49.17	88.63	87.99
3	−1(15)	1(70)	−1(20)	60.84	59.24	91.47	91.15
4	−1(15)	1(70)	1(40)	65.10	65.31	91.25	90.90
5	1(45)	−1(30)	−1(20)	44.48	44.09	88.38	88.12
6	1(45)	−1(30)	1(40)	45.84	47.25	88.06	87.77
7	1(45)	1(70)	−1(20)	50.24	52.97	90.64	90.67
8	1(45)	1(70)	1(40)	65.73	64.58	90.47	90.10
*Axial points*							
9	−α (4.77)	0(50)	0(30)	62.66	64.55	90.64	91.27
10	α (55.23)	0(50)	0(30)	59.30	57.67	90.46	90.69
11	0(30)	−α (16.36)	0(30)	35.42	35.88	86.76	87.31
12	0(30)	α (83.64)	0(30)	57.11	56.91	91.59	91.90
13	0(30)	0(50)	−α (13.18)	53.58	52.48	88.26	88.43
14	0(30)	0(50)	α (46.82)	58.87	60.24	87.24	87.93
*Central points*							
15 (C)	0(30)	0(50)	0(30)	60.70	60.43	93.27	93.71
16 (C)	0(30)	0(50)	0(30)	60.20	60.43	94.29	93.71

*Explanations*: X_1_, X_2_, and X_3_—independent variables, namely time of extraction (min), ethanol concentration (%), and solvent-to-raw-material ratio (mL/g dry wt.), respectively; TPC (total phenolic content) and %I (antioxidant activity)—dependent CCD parameters; GAE—gallic acid equivalent; * average of the triple determinations.

**Table 2 molecules-28-06335-t002:** Regression coefficients (*R*^2^) of the predicted second-order polynomial models for the total phenolic content (TPC) and antioxidant activity (%I) of MSL extracts.

Coefficient	TPC Model		%I Model	
	*R* ^2^	S.E.	*R* ^2^	S.E.
*Intercept*				
*β* _0_	24.60	2.10	58.95 ^a^	4.29
*Linear*				
*β* _1_	−0.53	0.05	0.29	0.09
*β* _2_	1.21 ^a^	0.04	0.45	0.08
*β* _3_	0.29	0.08	1.19	0.17
*Quadratic*				
*β* _11_	0.00	0.00	−0.00	0.00
*β* _22_	−0.01 ^a^	0.00	−0.00	0.00
*β* _33_	−0.01	0.00	−0.02	0.00
*Interaction*				
*β* _12_	0.00	0.00	−0.00	0.00
*β* _13_	0.01	0.00	−0.00	0.00
*β* _23_	0.01 ^a^	0.00	−0.00	0.00

*Explanations*: S.E.—standard error; ^a^ statistically significant (*p* < 0.05).

**Table 3 molecules-28-06335-t003:** Analysis of variance (ANOVA), including regression coefficients (*R*^2^) of the second-order polynomial models, related to total phenolic content (TPC) and antioxidant activity (%I) of MSL extracts.

Independent Variables	*SS*	*df*	*F*-Value	*p*-Value
**TPC**				
*Linear*				
X_1_	57.25	1	457.97	0.0297 ^a^
X_2_	534.13	1	4273.04	0.0097 ^b^
X_3_	72.69	1	581.49	0.0264 ^a^
*Quadratic*				
X_1_^2^	0.54	1	4.33	0.2853
X_2_^2^	228.00	1	1823.98	0.0149 ^a^
X_3_^2^	19.20	1	153.58	0.0513
*Interaction*				
X_1_X_2_	0.70	1	5.62	0.2542
X_1_X_3_	15.37	1	122.99	0.0573
X_2_X_3_	35.66	1	285.27	0.0376 ^a^
Lack of fit	32.52	5	52.03	0.1048
Pure error	0.13	1		
Total *SS*	1037.60	15		
*R* ^2^	0.9685			
*R*^2^_adj_.	0.9213			
**%I**				
X_1_	0.41	1	0.78	0.5396
X_2_	25.42	1	48.87	0.0905
X_3_	0.31	1	0.59	0.5833
X_1_^2^	8.60	1	16.54	0.1535
X_2_^2^	19.47	1	37.43	0.1031
X_3_^2^	35.36	1	67.96	0.0768
X_1_X_2_	0.17	1	0.33	0.6685
X_1_X_3_	0.05	1	0.10	0.8036
X_2_X_3_	0.03	1	0.05	0.8618
Lack of fit	2.35	5	0.90	0.6592
Pure error	0.52	1		
Total *SS*	69.06	15		
*R* ^2^	0.9585			
*R* ^2^ _adj_	0.8961			

*Explanations*: ^a,b^ statistically significant at *p* < 0.05 and *p* < 0.01, respectively; *SS*—sum of squares; *df*—degrees of freedom; _adj_—adjusted value.

**Table 4 molecules-28-06335-t004:** Polyphenolic compounds identified and quantified in MSL flower bud extracts while simultaneously using LC/ESI-QToF/MS-MS (in negative-ion mode) and RP-LC/PDA methods.

No.	Compound	R_t_ (min)	λ_max_ (nm)	Formula	Precursor Ion (*m*/*z*)	Product Ions (*m*/*z*)	Content (mg/g dry wt.)
1	Protocatechuic acid ^R^	3.67	206, 260, 294	C₇H₆O₄	153.0193	110.0331, 109.0297, 108.0219	0.47
2	Chlorogenic acid	4.58	218, 326	C_16_H_18_O_9_	353.0855 ^R^	191.0543, 173.0437	0.91
3	Echinacoside ^R^	5.08	198, 330	C_35_H_46_O_20_	785.2466	623.2127, 477.1619, 315.0898, 179.0338, 161.0233	2.52
4	2′-Rhamno- echinacoside	9.91	198, 330	C_41_H_56_O_24_	931.3058	769.2726, 751.2572, 179.0499, 161.0238	19.21
5	Vanillic acid ^R^	12.41	218, 260, 292	C_8_H_8_O_4_	167.0351	135.0116, 109.0250, 108.0210	0.26
6	Quercetin 3-*O*-neohesperidoside	19.34	204, 266, 350	C_27_H_30_O_16_	609.1303	300.0277, 271.0268, 151.0039	1.05
7	Quercetin 3-*O*-rutinoside (Rutoside) ^R^	20.95	204, 256, 355	C_27_H_30_O_16_	609.1443	301.0405, 300.0334, 271.0253, 257.0411, 229.0108	27.99
8	Quercetin 3-*O*-glucoside (Isoquercitrin) ^R^	24.37	204, 256, 355	C_21_H_20_O_12_	463.0889	301.0347, 271.0229, 178.9999	1.45
9	Acteoside (Verbascoside) ^R^	26.48	198, 330	C_29_H_36_O_15_	623.1985	461.1665, 179.0342, 161.0246	8.58
10	Kaempferol 3-*O*-rutinoside (Nicotiflorin) ^R^	28.89	196, 266, 346	C_27_H_30_O_15_	593.1571	345.0664, 285.0444	3.39
11	Isorhamnetin 3-*O*-rutinoside (Narcissin)	29.69	204, 254, 354	C_28_H_32_O_16_	623.1584	315.0516, 314.0441, 300.0296, 271.0252, 255.0216, 161.0243	1.25
12	Isorhamnetin 3-*O*-glucoside ^R^	30.80	204, 254, 354	C_22_H_22_O_12_	477.1042	315.0486, 314.0429, 271.0256	0.16
13	Isorhamnetin 3-*O*-rutinoside isomer	35.52	205, 255, 354	C_28_H_32_O_16_	623.1611	315.0505, 314.0437	0.95
14	Rhamnazin 3-*O*-rutinoside (Ombuoside)	38.44	206, 256, 356	C_29_H_34_O_16_	637.1728	330.0684, 329.0653, 315.0509, 288.0168, 161.0239	0.21
15	Magnolin ^R^	56.22	204, 230, 278	C_23_H_28_O_7_	415.4612	236.1059, 222.1580, 221.1545, 220.1469	0.12
16	Lignan (fargesin type)	58.93	202, 236, 260	n.d.	595.2865	279.2333, 174.9563, 112.9860	0.41
17	Lignan (fargesin type)	60.54	202, 234, 286	n.d.	571.2938	309.2091, 174.9570, 112.9856	0.47
18	Fargesin ^R^	62.25	202, 234, 284	C_21_H_22_O_6_	369.1328	357.1360, 242.9433, 174.9563, 112.9856	0.12
19	Lignan (fargesin type)	66.47	204, 234, 280	n.d.	293.2140	223.1358,195.1402, 174.9570, 112.9856	0.79
20	Lignan (fargesin type)	66.87	204, 236, 286	n.d.	625.3393	341.1096, 255.2333, 174.9561, 112.9856	0.62
21	Lignan (fargesin type)	67.37	202, 234, 286	n.d.	317.1745	274.1890, 174.9560, 112.9856	0.68
22	Lignan (fargesin type)	68.08	202, 234, 284	n.d.	295.2280	277.2182, 174.9564, 112.9856	0.71

*Explanations*: ^R^ identity of the compound additionally confirmed using a reference substance; n.d.—not determined.

**Table 5 molecules-28-06335-t005:** Experimental parameters and coded variables of central composite design (CCD).

		Coded Variables				
Independent Variables	Unit	−α	−1	0	1	+α
Time of extraction (X_1_)	Min	4.77	15	30	45	55.23
Ethanol concentration (X_2_)	%	16.36	30	50	70	83.64
Solvent-to-plant-material ratio (X_3_)	mL/g	13.18	20	30	40	46.82

*Explanations*: −α and +α indicate lower and higher axial values, respectively.

## Data Availability

Not applicable.

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
