# Peer review of "Response Surface Methodology in Optimising the Extraction of Polyphenolic Antioxidants from Flower Buds of Magnolia × soulangeana Soul.-Bod. var. ‘Lennei’ and Their Detailed Qualitative and Quantitative Profiling"

_molecules, 2023, doi:10.3390/molecules28176335_

Round 1

Reviewer 1 Report

The manuscript "Response surface methodology in optimizing the extraction of polyphenolic antioxidants from flower buds of Magnolia x soulangeana Soul.-Bod. var. 'Lennei' and their detailed phytochemical profiling using coupled chromatographic, spectroscopic, and tandem m", reports relevant information on compounds polyphenols of M. soulangeana with antioxidant properties. However, the authors should clarify the following points:

·        Among the lignans identified, the authors did not observe signs of neolignans?

·        the addition of chemical structures would help to understand the results.

·        How they solved the filtration step required for ultrasound-assisted extraction

·        With this novel technique, it is possible to obtain compounds degraded by the high frequencies used. In this sense, could some of the reported compounds be degradation products?

·        Some of the identified compounds may cause interference because they have an absorption spectrum similar to DPPH. In this sense, the authors could perform other experiments such as ABTS and DMPD, to corroborate the results obtained.

·        The reported compounds were identified as glycosides or aglycones. Could the frequencies used have broken the O-sugar bond?

The authors must correct the manuscript, since there are parts written in British English and parts written in American English.

Author Response

Dear Reviewer! Please see the attachment.

Reviewer 2 Report

This is an interesting and well-structured work, that uses response surface methodology to find the best conditions for polyphenols extraction from Magnolia x soulangeana.

This is relevant since there are few reports about the phytochemical content of Magnolia x soulangeana, and apparently none regarding processing with ultrasound-assisted extraction or other related technique.

Title: the title is too long. I suggest making it shorter. Generally, a good title is descriptive but not longer than 20 words

Abstract:

Elaborate on the novelty of RSM to set the best extraction conditions. Response surface methodology (RSM) is a widely-used statistical design for this purpose.

Introduction

Provide a concise explanation justifying why authors decided to study this plant. 

Highlight the gap in knowledge or novelty addressed with this work.

Why don’t the authors used other solvents to make the experimental design more robust? If different solvents are used, author can set the optimal conditions for polyphenol extraction.

The objective in the introduction section should be improved. Please, write the aim of the study in more detail. 

Methods.

This section is well-described.

L42: There is a typo: “cooridinates”

Section 3.1. Please include the certificate that shows the botanical identification.  

To enhance transparency, authors could add more details about how the plants were collected (weight, experimental units, replicates).

Results

Since there are no previous studies about the phenolic content of MSL, provide a brief justification for using these standards for LC.

L452 - Why the concentration range is so narrow for the standard curve?

Section 3.3 - Clarify the basis for selecting experimental parameters for CCD. Describe any preliminary experiments conducted to establish upper and lower limits for X1, X2, and X3. Justify the preference for the DPPH antioxidant test over alternative assays like ABTS, FRAP, and ORAC.

Please include the characteristics of the ultrasonic bath (power W/cm2))

L471 - Define "MFL."

L474 - Verify whether is the correct section

L475-477 - Explain the basis for these selected conditions.

L484 -Clarify whether the volume of MeOH added to the dry residue was 25 mL or if the tube's capacity was 25 mL

L485 - Indicate the difference between MFL and MSL

Results:

L40 - Clarify this sentence

L190 - Include the TPC content in this study.

L193 - Include the TPC yield in this study.

L188 and L204: Provide an explanation for the discrepancies between predicted and experimental values.

L207 - Include the antioxidant value in this study.

Consider comparing the optimal conditions for higher TPC extraction against antioxidant capacity for a more comprehensive conclusion.

Section 2.2

In general, this section is well structured and discussed.

L 213, 268, 293, 376, 396 - How can you be sure that all the 22 are antioxidants?

It would be valuable to show the quantification of individual phenolics, all those mentioned in section 3.2

Conclusions

Add a concise sentence summarizing the key findings of the study in terms of optimal extraction conditions.

Overall, the writing is comprehensible; I only suggest double-checking the grammatical structure and style.

Author Response

(The authors gave the same response as above.)
